# Histopathological and Immunological Findings in the Common Marmoset Following Exposure to Aerosolized SARS-CoV-2

**DOI:** 10.3390/v14071580

**Published:** 2022-07-21

**Authors:** Rachel E. Ireland, Carwyn D. Davies, Emma Keyser, James S. F. Findlay, Lin Eastaugh, Thomas R. Laws, Francisco J. Salguero, Laura Hunter, Michelle Nelson

**Affiliations:** 1CBR Division, Defence Science and Technology Laboratory (Dstl), Salisbury SP4 0JQ, UK; cdavies@dstl.gov.uk (C.D.D.); ekeyser@dstl.gov.uk (E.K.); jsfindlay@dstl.gov.uk (J.S.F.F.); leastaugh@dstl.gov.uk (L.E.); trlaws@dstl.gov.uk (T.R.L.); mnelson@dstl.gov.uk (M.N.); 2United Kingdom Health and Security Agency, Salisbury SP4 0JG, UK; javier.salguero@ukhsa.gov.uk (F.J.S.); laura.hunter@ukhsa.gov.uk (L.H.)

**Keywords:** SARS-CoV-2, aerosol, marmoset, ACE2, TMPRSS2

## Abstract

There is an enduring requirement to develop animal models of COVID-19 to assess the efficacy of vaccines and therapeutics that can be used to treat the disease in humans. In this study, six marmosets were exposed to a small particle aerosol (1–3 µm) of SARS-CoV-2 VIC01 that delivered the virus directly to the lower respiratory tract. Following the challenge, marmosets did not develop clinical signs, although a disruption to the normal diurnal temperature rhythm was observed in three out of six animals. Early weight loss and changes to respiratory pattern and activity were also observed, yet there was limited evidence of viral replication or lung pathology associated with infection. There was a robust innate immunological response to infection, which included an early increase in circulating neutrophils and monocytes and a reduction in the proportion of circulating T-cells. Expression of the ACE2 receptor in respiratory tissues was almost absent, but there was ubiquitous expression of TMPRSS2. The results of this study indicate that exposure of marmosets to high concentrations of aerosolised SARS-CoV-2 did not result in the development of clear, reproducible signs of COVID-19.

## 1. Introduction

The first cases of COVID-19, caused by severe acute respiratory syndrome coronavirus 2 (SARS-CoV-2) emerged in Wuhan, Hubei Province, China in December 2019 and since then, SARS-CoV-2 infections have spread rapidly around the globe. Infection usually results in a mild flu-like illness that includes fever, fatigue, dry cough often with olfactory dysfunction, including anosmia [1,2]. However, in some cases, disease can progress to severe pneumonia, acute respiratory distress syndrome (ARDS), multi-organ failure, and even death [3,4,5,6]. Individuals at a greater risk of developing severe COVID-19 include those with underlying health conditions such as obesity, diabetes, chronic respiratory disease, and cardiovascular disease [7,8]. A worldwide effort to combat this disease has resulted in the successful deployment of numerous vaccine candidates and therapeutics, however the COVID-19 pandemic is presenting new challenges with the emergence of new variants of concern (VOCs) with possible enhanced transmission and immune evasion [9,10].

A number of animal species have been used to develop models of COVID-19, and these models have played a vital role in the preclinical development of novel vaccine candidates and therapeutics prior to use in humans (reviewed in [11]). These models are essential in the continued research effort to address challenges posed by new SARS-CoV-2 variants, in particular the virulence, transmission, and immune escape. Despite these research efforts, animal models that can recapitulate key features of the severe and lethal form of disease in humans has been identified as a major gap in the armory of SARS-CoV-2 infection models available for preclinical studies (reviewed in [12]). Therefore, there is an enduring requirement to develop and characterize alternative models that would fulfil this need.

The common marmoset may be a suitable model of human COVID-19 disease, as marmoset models have previously been developed for highly pathogenic severe acute respiratory syndrome coronavirus (SARS-CoV-1) [13] and Middle Eastern Respiratory Syndrome coronavirus (MERS-CoV) [14,15,16], with the disease closely resembling features of disease in humans. In response to the SARS-CoV-2 pandemic, several groups have undertaken COVID-19 modelling with marmosets using either the intranasal route only or a combination of intranasal, intratracheal, and ocular routes of infection [17,18]. No clinical signs or virus shedding were observed when marmosets were challenged by the intranasal route only; however, virus was detected in the blood, throat, and nose of animals challenged by multiple routes.

This study aimed to uniquely assess the susceptibility of the marmoset to aerosolised SARS-CoV-2. This challenge route delivers aerosols containing the virus into the deep lung and has the potential to cause a more severe form of the disease, and possibly even at lower challenge doses [16,19]. This increases the likelihood of marmosets developing clear, reproducible signs of disease that are relevant to human COVID-19. This study also aimed to support the principles of the 3Rs by increasing the knowledge of SARS-CoV-2 infection in the marmoset, in particular using continuous monitoring of core body temperature (Tc) to assess the fever response, an extensive immunological assessment, and the establishment of immunohistochemistry and RNAScope in situ hybridization methodologies to detect viral RNA, ACE2, and TMPRSS2 in marmoset tissues.

## 2. Materials and Methods

### 2.1. Cell Lines

Vero/hSLAM cell line (04091501, ECACC, Salisbury, UK) and Vero C1008 cell line (85020206, ECACC, Salisbury, UK) were maintained in Dulbecco’s minimal essential media (DMEM) supplemented with 2 mM L-glutamine, 100 U/mL penicillin, 100 μg/mL streptomycin, and 10% (*v*/*v*) foetal calf serum (FCS) (all Sigma-Aldrich, Gillingham, UK) at 37 °C in 5% (*v*/*v*) CO_2_ humidified atmosphere. For cell infections, cells were maintained in Leibovitz’s L-15 medium supplemented 2 mM L-glutamine, 100 U/mL penicillin, 100 μg/mL streptomycin, and 2% (*v*/*v*) FCS (all Sigma-Aldrich, Gillingham, UK) at 37 °C without CO_2_ in a humidified atmosphere. Vero/hSLAM cells were also supplemented with 0.4 μg/mL Geneticin^®^ (10131027, Gibco^™^ Thermo Fisher Scientific, Paisley, UK).

### 2.2. Viruses, Propagation and Enumeration

SARS-CoV-2 (BetaCoV/Australia/VIC01/2020 (Passage 2)) was obtained from the Victorian Infectious Diseases Reference Laboratory at the Peter Doherty Institute for Infection and Immunity in Melbourne, Australia [20]. A Master stock (Passage 3) and Working stocks (Passage 4) were propagated in Vero/hSLAM cell line. To generate high titre stocks for aerosol challenge, the virus was first concentrated using Amicon^®^ Ultra-15 100 k centrifugal filter units for 25 min at 4000× *g*, at 4 °C (UFC910024, Merck Millipore, Watford, UK) prior to a final purification step through a 30% (*w*/*v*) sucrose cushion at 179,200× *g* for 2 h, at 4 °C. Genome stability was verified following the passage using an amplicon-based Illumina whole genome sequencing approach with the Liverpool protocol/primers [21]. Sub-consensus variant analysis was conducted using LoFreq [22]. Virus was enumerated using plaque assay or Reed & Muench TCID_50_ method [23] in Vero C1008 cells, as indicated. The presence of replicating virus was assessed in the lungs, liver, spleen, kidney, brain, and blood by plaque assay and a further blind passage in Vero C1008 cells for 7 days.

### 2.3. Animals

Healthy, sexually mature common marmosets (*Callithrix jacchus*) were obtained from the Dstl Porton Down breeding colony and housed as female and vasectomised male pairs. The overall weight of animals when assigned to the study ranged from 370 to 440 g (mean of 406.2 ± 25.33 g). The overall age ranged from 3.7 to 4.7 years (mean of 4.3 ± 0.4 years). Animals were randomly assigned (in pairs) to each group. Animals were allowed free access to food and water as well as environmental enrichment throughout the study. All animals were surgically implanted intraperitoneally with a Remo 201 device (EMMS, Bordon, UK) under general anaesthesia (ketamine hydrochloride/medetomidine hydrochloride/isofluorane in oxygen) to record core body temperature (Tc). Data were transmitted from the devices at 30 s intervals to locally placed antennas and relayed to receivers. Data were analyzed using the eDacq software (v1.9.1 BETA 3, EMMS, Bordon, UK) to provide real-time and recordable Tc. The animal studies were carried out in accordance with the UK Animals (Scientific Procedures) Act of 1986 and the Codes of Practice for the Housing and Care of Animals used in Scientific Procedures 1989. Following a challenge with SARS-CoV-2, animals were handled under animal containment level 3 (CL3) conditions, within a half-suit isolator compliant with British Standard BS5726.

### 2.4. Aerosol Conditions Optimisation

Prior to undertaking in vivo studies, the conditions for aerosolization of the virus were optimized at different relative humidities (~35%, ~50%, and ~80%) on 3 separate occasions. Two concentrations of the virus diluted in tissue culture fluid supplemented with 2% FCS (1 × 10^7^ to 1 × 10^8^ TCID_50_/mL) were used to load separate Collison nebulisers. The aerosol system was conditioned to the appropriate relative humidity. To mimic a 10 min animal exposure, an all-glass impinger (AGI-30; Ace Glass Inc., Vineland, NJ, USA) was collected between 4.5 and 5.5 min (i.e., the mid-point of exposure). The impinger contained 10 mL of tissue culture fluid supplemented with 2% FCS and the aerosol cloud was sampled for 1 min. The flow rate of the collection impinger was fixed at 12 L/minute. An Aerodynamic Particle Sizer (APS) (Model 3321, TSI Incorporated, Shoreview, MN, USA) equipped with a 1:100 diluter (Model 3302A, TSI Incorporated, Shoreview, MN, USA) was included into the system and sampled the aerosol cloud for 1 min at 2.5 and 6.5 min from the start of aerosolization.

The starting concentration of the virus (C_neb_), the concentration of virus recovered by impingement (C_samp_) and the Spray factor (S_f_) was determined and compared for each of the conditions. The mass median aerodynamic diameter (MMAD) and the geometric standard deviation (GSD) were calculated using Aerosol Instrument Manager^®^ v9.0 (TSI Incorporated, Shoreview, MN, USA) and were compared between humidities.

### 2.5. Marmoset Challenge

Prior to challenge, pairs of animals were sedated with 10 mg/kg of ketamine hydrochloride delivered intra-muscularly. Animals were secured within plethysmography tubes and placed within the exposure unit. Briefly, a 4.5 mL suspension of 1.6 × 10^8^ TCID_50_/mL of SARS-CoV-2 VIC01 (C_neb_) was aerosolized using a 3-jet Collison nebuliser in conjunction with an AeroMP (Biaera Technologies L.L.C., Hagerstown, MD, USA) [24]) at an RH of 43.4% ± 1.1%. Animals were exposed to aerosolized virus for a period of ten minutes and sampled to assess the concentration of the virus and the particle size as described above. The virus was enumerated using TCID_50_ assay. The presented challenge dose was calculated for each animal using the concentration of the virus in the aerosol and real-time plethysmography to determine the volume of air breathed by the animal using eDacq software (v1.9.1 BETA 3, EMMS, Bordon, UK). The concentration of the virus in the aerosol cloud was calculated as described below:(1)$$\mathrm{Aerosol}\;\mathrm{Concentration}\;\left({\mathrm{TCID}}_{50}\;\mathrm{per}\;L\;\mathrm{of}\;\mathrm{air}\right)=\frac{\mathrm{Impinger}\;\mathrm{count}\left({\mathrm{TCID}}_{50}\;\mathrm{per}\;\mathrm{mL}\right)\;\times \;\mathrm{Impinger}\;\mathrm{volume}\left(\mathrm{mL}\right)}{\mathrm{Impinger}\;\mathrm{flow}\;\mathrm{rate}\left(L\;\mathrm{per}\;\mathrm{minute}\right)\;\times \;\mathrm{Impinger}\;\mathrm{time}\left(\mathrm{minutes}\right)}$$
(2)$$\mathrm{Dose}\;\mathrm{received}\;\left({\mathrm{TCID}}_{50}\right)=\mathrm{Aerosol}\;\mathrm{concentration}\;\left({\mathrm{TCID}}_{50}\;\mathrm{per}\;L\;\mathrm{of}\;\mathrm{air}\right){\;\times \;\mathrm{Total}\;\mathrm{accumulated}\;\mathrm{volume}\;(L)}^{*}$$* Obtained using real-time plethysmography for each animal, sampled during exposure.

The spray factor (S_f_) (ratio of the concentration of virus per L of air and the starting concentration in the Collison nebuliser) was calculated as described below and used as a measure of the reproducibility of the aerosol system and to compare aerosolization conditions.
(3)$$\mathrm{Spray}\;\mathrm{Factor}\;(\mathrm{Sf})=\frac{{\mathrm{Aerosol}\;\mathrm{concentration}\;(\mathrm{TCID}}_{50}\;\mathrm{per}\;L\;\mathrm{of}\;\mathrm{air})}{{\mathrm{Concentration}\;\mathrm{of}\;\mathrm{virus}\;\mathrm{in}\;\mathrm{the}\;\mathrm{nebuliser}\;(\mathrm{TCID}}_{50\;}\mathrm{per}\;\mathrm{mL}\;\times 1000)}$$

### 2.6. Post-Challenge Observations

The animals were observed at least 3 times per day after the challenge, for up to 21 days. Clinical signs were scored during physical entry into the room and Tc were recorded during silent hours for each animal. Animals were conditioned to weighing prior to being assigned to the study using a balance placed within their home cage, thus avoiding the need to handle the animal to obtain this data. The animals were incentivized to sit in a weighing bucket containing a “treat” that is placed on the balance. Body weight data was collected weekly during the pre-containment phase of the study. Once in high containment, body weight data was collected daily in the 5 days prior to the challenge and continued daily throughout the study. This method of weighing relies on animal co-operation and therefore there is an occasional data point missing. On days where procedures were conducted on marmosets, body weight data was collected whilst the animals were anaesthetized.

### 2.7. In-Life Data Collection

The schedule for in-life data collection is summarized in Figure 1. Briefly, on days −14, 1, 2, 3, 4, 7, 14, and 21 post-challenge, groups of animals were anaesthetized with a cocktail of fentanyl (0.01 mg/kg), medetomidine (0.06 mg/kg), and midazolam (0.5 mg/kg) (FMM) delivered intra-muscularly. At the end of sampling, anaesthesia was reversed using a cocktail of naloxone (0.005 mg/kg), flumazenil (0.1 mg/kg), and atipamezole (0.3 mg/kg) (NFA) delivered intra-muscularly. Anaesthetic doses were calculated according to the individual animal weight. Whilst sedated, blood was collected from the femoral vein into sodium citrate blood tubes for viral load assessment and immunological analysis. Nasal and throat swabs were collected to assess virus shedding using a flocked nasopharyngeal sampling swab into HiViral™ transport medium (VTM) (A04-96000T and AL167CS-50NO; both supplied by Trafalgar Scientific, Leicester, UK) and vortexed prior to storage at −20 °C.

### 2.8. Study Termination and Post-Mortem Sampling

The schedule for post-mortem sampling is summarized in Figure 1. Briefly, pairs of marmosets were euthanised at day 2, 4, or at the end of the study on day 21 post-challenge, using an overdose of sodium pentobarbitone administered intraperitoneally. Blood was collected by cardiac puncture into sodium citrate blood tubes for viral load assessment and immunological analysis. Throat and nasal swab samples were collected as described above. The liver, spleen, kidney, brain, gastrointestinal tract, thymus, and mediastinal lymph node were removed. In addition, the whole head and the respiratory tract (trachea and whole lung) were removed. Sections of each organ were collected for processing for viral load and immunological assessment, and the remainder of the respiratory tract, liver, spleen, kidney, gastrointestinal tract, mediastinal lymph node, and the whole head were collected into 10% (*v*/*v*) neutral buffered saline for histopathological assessment.

### 2.9. Detection of Viral RNA Using RT-PCR

RT-PCR was used to determine the presence of viral RNA in representative organs including lungs, liver, spleen, kidney, brain, and blood and in throat and nasal swab samples. Viral RNA was extracted from blood and tissue homogenates using QIAamp^®^ Viral RNA mini kit (52904, Qiagen, Manchester, UK). Viral RNA was extracted from throat and nasal swabs using the MagMax Pathogen RNA/DNA kit, Kingfisher Flex (4462359, Thermo Fisher Scientific, Paisley, UK). Real-time RT-PCR assays for the SARS-CoV-2 *E gene* were performed in duplicate using the method developed by Corman et al. [25], Briefly, 5 µL of RNA sample was added to 20 µL of TaqMan™ Fast Virus 1-Step Master Mix (4444432, Thermo Fisher Scientific, Paisley, UK) containing primers E_Sarbeco_F ACAGGTACGTTAATAGTTAATAGCGT (400 nM per reaction), E_Sarbeco_P1 FAM-ACACTAGCCATCCTTACTGCGCTTCG-BBQ (200 nM per reaction), E_Sarbeco_R ATATTGCAGCAG TACGCACACA (400 nM per reaction). Thermal cycling was performed at 55 °C for 10 min for reverse transcription, followed by 94 °C for 3 min and then 45 cycles of 94 °C for 15 s, 58 °C for 30 s.

### 2.10. Histopathology

Sections of lung, nasal cavity (longitudinal section close to the medial septum after decalcification with 10% formic acid solution), trachea, liver, kidney, spleen, mediastinal lymph node, thymus, oesophagus, small, and large intestine from each animal were fixed in 10% neutral-buffered formalin (for 16 to 35 days), processed to paraffin wax and 4 µm thick sections cut and stained with haematoxylin and eosin (H&E). The tissues specified above were examined by light microscopy and evaluated subjectively by a qualified pathologist. Tissue sections were scanned by a Hamamatsu NanoZoomer S360 and viewed with NDP.view2 software (v2.8.24, Hamamatsu, Japan). The pathologist was blinded to the group details and the slides were randomized prior to examination in order to prevent bias (blind evaluation).

### 2.11. SARS-CoV-2 RNA Staining by ISH (RNAScope)

RNAScope^®^ was used to determine the presence of SARS-CoV-2 virus in tissues. Briefly, tissues were pre-treated with hydrogen peroxide and incubated for 10 min at room temperature, with target retrieval for 15 min at 98–101 °C followed by protease plus treatment for 30 min at 40 °C (322350, Advanced Cell Diagnostics, Newark, NJ, USA). A V-nCoV2019-S probe (848561, Advanced Cell Diagnostics, Newark, CA, USA) targeting the *S gene* was incubated on the tissues for 2 h at 40 °C. Amplification of the signal was carried out following the RNAScope protocol (RNAScope 2.5 HD Detection Reagent—Red) using the RNAScope 2.5 HD red kit (322350, Advanced Cell Diagnostics). Positive control sections (Rhesus macaque lung from experimentally infected animals with SARS-CoV-2) and negative controls were used in the RNAScope ISH runs.

### 2.12. ACE2 and TMPRSS2 Distribution (IHC)

Tissue sections from each animal were cut at 4 µm on superfrost slides and stained using immunohistochemistry to detect ACE2 and TMPRSS2. The Leica Bond Rxm and polymer refine detection kit with HRP (DS9800, Leica Biosystems, Wetzlar, Germany) were used to visualize the antigens. Sections were dewaxed, rehydrated, and treated in 3–4% hydrogen peroxide for 5 min to quench endogenous peroxidase activity. Anti-ACE2 rabbit monoclonal antibody (NBP267692, Novus Biologicals, Centennial, CO, USA) and TMPRSS2 rabbit polyclonal antibody (NBP293322, Novus Biologicals) were used at dilutions of 1:250 and 1:1000, respectively, and were incubated for 30 min. Leica polymer refine detection kit (DS9800, Leica Biosystems) was used, and DAB as chromogen for visualization and sections were counterstained with Harris’ haematoxylin (DS9800, Leica Biosystems). Positive control sections (Rhesus macaque lung and jejunum experimentally infected with SARS-CoV-2) and negative controls were used in the IHC runs. Images were scanned digitally using a Hamamatsu S360 digital slide scanner and examined using NDP.view2 software (v2.8.24, Hamamatsu, Japan).

### 2.13. Cell Phenotyping

Blood, spleen, and lung aliquots were stained for an array of immunological markers. Blood samples (collected into sodium citrate and further diluted 50:50 with PBS) were centrifuged at 300× *g* for 5 min, and the plasma was stored frozen at −70 °C for later analysis. The cell pellets were lysed in 2 mL of red cell lysis buffer (555899, BD biosciences, Oxford, UK) for 5 min then centrifuged at 300× *g* for 5 min. White cells were re-suspended in zombie UV stain (423107, BioLegend, San Diego, CA, USA) for 5 min, followed by the addition of 100 µL cell staining mix, incubated at room temperature for 30–40 min whilst protected from light. Stain mix comprised of fluorescently bound anti-human or anti-marmoset antibodies: CD45 (6C9: 250204), CD20 (Bly1: Sc-19990), TCR γδ (B1: 331220), CD27 (M-T721: 564894), CD279 PD-1 (EH12.2E7: 329930), CD163 (GHI/61: 333624), CD80 (2D10: 305236), MHCII (L243: 307630), CD40 (5C3: 334336), CD16 (3G8: 302018), CD64 (10.1: 305020), CD54 (HCD54: 322716) all from BioLegend (San Diego, CA, USA); antibodies: CD3 (SP34-2: 741872), CD8 (HIT8a: 740303), CD56 (B159: 741375), CD69 (FN50: 750213), CD14 (M5E2: 557742), CD11c (SHCL3: 340544) from BD biosciences (Oxford, UK).

Cells were washed with 1 mL of PBS followed by centrifugation at 300× *g*, and the cell pellet was re-suspended in 4% (*v*/*v*) paraformaldehyde for 36 h at 4 °C. Cell populations were measured by flow cytometry on an Aurora Cytek (Fremont, CA, USA) and analyzed using Flow Jo v10 (BD, Franklin Lakes, NJ, USA). The CD40 staining for day 2 samples failed, therefore this data was not included in the final analysis.

Lung and spleen samples were minced into 2 mL of L-15 media and gently stroked though a 45 µm cell sieve to obtain a viable single cell suspension. Spleen cells (50 µL) and lung cells (200 µL) were processed and stained as above.

### 2.14. Recall Assays

Spleen cells or white blood cells (separated as described above) were diluted to achieve a density of 1–3 × 10^6^ cells/mL, and stimulated with either L-15 media (negative control), ConA 2.5 µg/mL (positive control; C0412, Sigma-Aldrich, Gillingham, UK), or SARS-CoV-2 spike peptide mix (0.1 µg/mL Peptivator 130-126-700, Miltenyi Biotec, Surrey, UK) for 18 h incubated at 37 °C. The supernatant was removed and stored at −80 °C prior to cytokine analysis. To the cells, 1 µL/mL of brefeldin A (555029, BD biosciences, Oxford, UK) was added, and cells were re-incubated for 4 h. Cells were than harvested, stained for CD3^+^ T cells, CD8^+^ T cells, and CD56^+^ cells, and fixed in 4% (*v*/*v*) paraformaldehyde for 36 h at 4 °C. Following fixation, cells were permeabilized (561651, BD biosciences) and stained for intracellular IFN-γ (1-D1K 3420-7, Mabtech, Nacka Strand, Sweden) and measured by flow cytometry on an Aurora Cytek.

### 2.15. Cytokine Analysis

Cytokines and chemokines were measured in the plasma or lung and spleen homogenates that were stored frozen at −80 °C until required. Levels of cytokines and chemokines were quantified using the human flexset for IL-1β (558279), IL-6 (558276), MCP-1 (558287), and RANTES ((558324, all BD biosciences, Oxford, UK) and for TNF-α (antibody pair from kit CT772 U-CyTech, Ultrecht, The Netherlands) and IFN-γ (Antibody pair 1-D1K 3420-7 and MT126L:3421M-3 from Mabtech, Nacka Strand, Sweden, conjugated to flex beads by BBI Detection Ltd., Salisbury, UK). All samples were fixed in 4% paraformaldehyde for 36 h at 4 °C and analyzed by flow cytometry (FACS Canto II BD, Oxford, UK). The concentration of IFN-γ (3421M, Mabtech, Nacka Strand, Sweden) and C-reactive protein (CRP; ab260062, Abcam, Cambridge, UK) in plasma were measured by ELISA following manufactures instructions.

### 2.16. Antibodies

Total IgG was measured in plasma samples collected at post-mortem by ELISA using the Human SARS-CoV-2 Spike (Trimer) Ig Total ELISA Kit (BMS2323, Thermo Fisher Scientific, Paisley, UK) with the human IgG conjugate (AP112P, Sigma-Aldrich, Gillingham, UK), as per manufacturer’s protocol. Briefly, 10 µL of diluted plasma (1:10) was added to the well and incubated for 30 min at 37 °C. Standards and control were also included on the plate. The wells were washed three times using wash buffer before adding 100 µL of HRP-conjugate solution, and was incubated for 30 min at 37 °C. The wells were washed three times using wash buffer, then 100 µL substrate solution was added. The plate was incubated for a further 15 min at room temperature, then 100 µL of stop solution was added to each well. The optical density at 450 nm was determined for each well.

### 2.17. Plaque Reduction Neutralization Test (PRNT)

Vero C1008 cells were seeded into 24-well plates at a density of 1 × 10^5^ to 5 × 10^5^ cells/mL in Dulbecco’s minimal essential media (DMEM) supplemented with 2 mM L-glutamine, 100 U/mL penicillin, 100 μg/mL streptomycin, and 10% (*v*/*v*) foetal calf serum (FCS) (all Sigma-Aldrich, Gillingham, UK) and incubated at 37 °C in 5% (*v*/*v*) CO_2_ humidified atmosphere for 1 to 3 days. On the day of infection, the virus was diluted to a final concentration of 1200 pfu/mL in Leibovitz’s L-15 medium supplemented 2 mM L-glutamine, 100 U/mL penicillin, 100 μg/mL streptomycin, and 2% (*v*/*v*) foetal calf serum (FCS) (all Sigma-Aldrich, Gillingham, UK). Marmoset sera was diluted 1:5 and 1:10 in L-15 medium and was either heat treated at 56 °C for 30 min to inactivate complement or left untreated, then incubated with 250 µL of diluted virus for 1 h at 37 °C. A volume of 200 µL of virus–serum mixtures were transferred to seeded plates in duplicate onto Vero C1008 monolayers and allowed to adsorb at room temperature for 1 h, with occasional rocking. Then, 1 mL of carboxymethylcellulose (CMC) overlay media was added to each well, and plates were incubated for 7 days at 37 °C without CO_2_. Cells were fixed to a minimum final concentration of 1% (*v*/*v*) formaldehyde and stained with 0.12% (*w*/*v*) crystal violet solution to visualize plaques for counting.

### 2.18. Statistical Analysis

All statistical analysis was performed using GraphPad Prism version 8.0.1. (GraphPad Software, San Diego, CA, USA). Comparison of the spray factor and particle size was performed on logarithmically transformed data by one-way ANOVA analysis. The comparison of the effect of RH on the MMAD was performed on untransformed data by one-way ANOVA analysis. Linear regression of data was used to assess the relationship between the C_samp_ (impinger) and C_neb_ (Collison) and Spray Factor and C_samp_. A mixed-effects model was used to determine statistical difference in weight change with time. Statistical differences were determined using ordinary one-way ANOVA with Tukey’s multiple comparisons or Dunnett’s multiple comparisons test on log transformed data (Y = LogY) for clinical chemistry and haematological parameters.

Ensuring consistent volumes of blood collected as small bleed samples obtained from in-life animals was difficult. Therefore, statistical differences between cell types were only performed on relative cell proportions rather than total count data, although for clarity this is also quoted. Neutrophil and monocyte/macrophage proportions and T:B ratios were determined by one-way ANOVA.

## 3. Results

### 3.1. Aerostability of SARS-CoV-2

Prior to performing in vivo studies, the aerosolization conditions for the virus were optimized. The efficiency of recovery of the virus at low (36.5 ± 4.4%), medium (52.3 ± 2.9%), and high (80.1 ± 3.3%) relative humidity (RH) was determined (Figure 2A). The particle size distribution was consistent at all three RH levels with the MMAD of the particles ranged from 0.93 to 1.30 µm (mean of 1.11 ± 0.09 µm,) with a GSD between 1.54 and 1.66 (Figure 2B,C). Linear regression of data was used to assess the relationship between the C_samp_ (impinger) and C_neb_ (Collison) (Figure 2D). The analysis indicated a highly probable positive relationship between these factors (*p* < 0.001, in all cases). Moreover, there was no evidence that the slopes were different between RH groups (*p* = 0.3204). Finally, there was no strong evidence for differences in the relationship between C_samp_ and C_neb_ that were related to RH (*p* = 0.0827). Unsurprisingly, this confirms that changes in the starting concentration (C_neb_) affects the concentration collected in the impinger (C_samp_) and by default the dose an animal would receive. This analysis also confirms that this is not affected by the RH.

The Spray Factor was used as a measure of the reproducibility of the aerosol system where the mean Spray Factor for low RH was 4.0 × 10^−6^, for medium RH was 7.51 × 10^−7^, and for high RH was 5.84 × 10^−7^. A linear regression was performed to determine whether there was a relationship between Spray Factor and C_samp_ (Figure 2E). For all RH subgroups, there was no relationship in the slope (*p* > 0.1 for all) indicating that the Spray Factor did not change significantly with the concentration of virus recovered in the impinger. However, the Y intercept was significantly different for each RH (*p* = 0.0024), suggesting better recovery of the virus with low RH than medium or high RH. Additionally, ANOVA comparisons indicate significantly better Spray Factor at low RH compared to both medium (*p* = 0.0266) and high RH (*p* = 0.0022), further suggesting more effective recovery of virus with low RH (Figure 2F). Therefore, an RH of between 40 to 50% was selected for the in vivo studies.

### 3.2. Marmosets Did Not Develop Clinical Signs of Infection Following Challenge with Aerosolized SARS-CoV-2

The aim of this study was to assess the susceptibility of marmosets to aerosolized SARS-CoV-2. Animals received a mean presented dose of 8.7 × 10^4^ TCID_50_ (±2.6 × 10^4^ TCID_50_) of SARS-CoV-2 VIC01 by the aerosol route. During the post-challenge period, clinical signs or signs of respiratory dysfunction were not observed in any animal during the course of the study. Animals did not develop fever; however, minor changes in core body temperature were observed between days 2 and 5 post-challenge in three out of the five animals that had continuous temperature monitoring, compared to the pre-challenge baseline (Figure 3). Note that there is no temperature data available for M2 as the temperature device failed. Animals M3, M4, and M5 exhibited an elevation in night time temperature that was greater than three standard deviations (SD) than the expected temperature, based on the pre-challenge baseline temperature (Figure 3B–D). Disruption to the temperature profiles were short-lived, with animals’ temperatures returning to normal within 30 h. Early changes in body weight were observed in M3, M4, M5, and M6 with up to 3% body weight loss within the first few days following SARS-CoV-2 challenge. Although the rate of daily weight change was not different during the post-challenge period (*p* = 0.401), body weight was significantly increased by day 21 post-challenge for animals M5 and M6 (*p* = 0.007), indicating recovery following infection (Figure 3F).

### 3.3. Viral RNA Was Detected in the Upper and Lower Respiratory Tract Following Initial Infection and There Was Limited Evidence of Viral Replication or Dissemination

Virus shedding was assessed using RT-PCR for the SARS-CoV-2 *E-gene* in both nasal and throat swabs throughout the study, as detailed Figure 1. Viral RNA was detected in nasal swabs from marmoset M6 until day 7 post-challenge, with a moderate positive PCR signal on day 1 post-challenge (Ct value of 25.30), and weak positive PCR signals on days 3, 4, and 7 post-challenge (Ct values of 30.77, 37.89, and 39.15 respectively). A weak positive PCR signal was detected in nasal and/or throat swabs from the remaining animals on day 1 post-challenge (Ct values > 30.77); however, viral RNA was not detected in these animals after this time (Ct values could not be determined). Replicating virus was not recovered from tissues or blood samples post-mortem by plaque assay. Using RT-PCR for the *E-gene*, viral RNA was detected in the lungs of animals on days 2 and 4 post-challenge (Ct values > 30.32); however, viral RNA was not detected on day 21 post-challenge (Ct values could not be determined), and viral RNA was not detected in any other tissues assessed (Ct values could not be determined). Replicating virus was not recovered from blood samples at any of the time points assessed by plaque assay and sterility tests also confirmed the absence of replicating virus. Due to low blood volumes collected, RT-PCR could not be conducted on these samples.

### 3.4. Immunological Response to Infection

#### 3.4.1. Increases in Proportions of Circulating Neutrophils and a Decrease in a Marmoset General Welfare Marker Indicates a Robust Innate Response to the Virus

Baseline levels of various immune cell phenotypes and genotypes were assessed in blood samples collected prior to infection. Following the challenge, blood was collected on day 1, 2, 3, 4, 7, 14, and 21 and compared to baseline levels. A significant shift in the proportions of neutrophils, monocytes, T-, and B-cells was observed over time (Figure 4). The average neutrophil density in the blood increased threefold from 3.1 × 10^4^/µL (baseline) to 9.54 × 10^4^/µL on day 21 post-challenge. The change in the proportion of neutrophils was most striking and significantly increased until day 3 post-challenge (*p* < 0.01 on day 1 and 3; *p* < 0.001 on day 2; Figure 4A). The proportions of neutrophils decreased on day 4, although they were still significantly higher than baseline values (*p* < 0.01), before further declining to non-significant proportions on day 7. A further increase in proportions were observed in the two remaining animals on day 14 and 21 (*p* < 0.001 in both cases). The expression of the neutrophil markers was also assessed. Generally, there was no change in expression of CD54^+^ (migration marker), CD16^+^ (human maturity marker), or CD64^+^ (sepsis indicator). However, greater than a 10% reduction in expression of HLA-DR^+^ (reported as a general indicator of marmoset health in Ngugi et al. [26]) was observed in three out of six animals between days 1 and day 7 (M3, M4, and M6) (Figure 4B). A reduction in circulating immature neutrophils was observed (high CD45^+^, low CD14^+^, smaller size), which is closely linked with innate responses to an infection (data not shown).

#### 3.4.2. There Was an Increase in Circulating Monocytes with Minimal Changes in Cell Expression Markers

Significant increases in circulating monocytes were also observed in all animals on day 1 post-challenge (*p* < 0.01), with levels returning to within the normal range by day 7 post-challenge (Figure 4C). There was also an increase in the number of monocytes expressing the classical activation marker CD40^+^ on day 1 compared to day 14 and 21 (*p* < 0.01), but not compared to pre-challenge baseline levels. There was no change in the expression of the inflammatory markers CD80^+^ (inflammatory marker), or CD16^+^ or CD163^+^ (alternative activation markers) (data not shown).

#### 3.4.3. Disruption in Normal Levels of Lymphocytes Were Observed despite the Lack of T-Cell Activation

Circulating T-cells were reduced in marmosets following the challenge with SARS-CoV-2. The average T cell density decreased nearly eightfold from 5 × 10^4^/µL (baseline) to 6.1 × 10^3^/µL on day 21 post-challenge, whereas the B cells remained relatively unaffected at 1.5 × 10^4^/µL (baseline) compared to 9.6 × 10^3^/µL on day 21 post-challenge. This resulted in a significant reduction in the T:B cell ratio observed at all time points assessed, compared to baseline (*p* < 0.01 on day 7, *p* < 0.001 on the remaining days; Figure 4D). There was no evidence of activation of the T-cells as measured by the CD4^+^, CD8^+^ cell ratio or CD3^+^:γδ^+^ T-cell ratios, or increased expression of PD-1 (programmed cell death) or CD56^+^ (cytotoxic activation) or reduced expression of CD27^+^ (early memory). There was limited evidence of early T-cell activation (CD69^+^); however, this was only significant on day 2 post-challenge (*p* < 0.02). There was no evidence of B-cell maturation with no changes in CD27^+^, HLA-DR^+^, and CD69^+^, by days 14 and 21 post-challenge.

#### 3.4.4. Macrophage Activation and Increasing Levels of Neutrophils Were Observed in the Lungs

The proportion of different immune cell types was also assessed in the lungs sampled on days 2, 4, and 21 post-challenge and compared to four naïve lung samples that were also processed during this study. A reduction in macrophages in the lungs was observed following challenge with SARS-CoV-2, and this was significantly reduced on day 21 compared to naïve lung samples (*p* < 0.05; Figure 4E). Despite the reduction in macrophage proportions, there was an increase in CD16^+^, CD80^+^, CD56^+^, CD69^+^, and CD40^+^ expression and a reduction in CD163^+^ expression on day 21 suggesting a level of classical macrophage activation (data not shown). An influx of neutrophils into the lungs was observed following the challenge with SARS-CoV-2 that was significantly increased on day 2 compared to naïve lung samples (*p* < 0.05) and declined with time (Figure 4E). There was no change in the T:B cell ratio in the lungs following challenge and there was no evidence of either T or B cell activation.

#### 3.4.5. Changes in Levels of Cytokine and Chemokines Were Not Observed

The level of IFN-γ, IL-1β, IL-6, MCP-1, and RANTES in blood and tissues of animals collected post-challenge did not differ from pre-challenge baseline levels (data not shown). IFN-γ was also measured by a more sensitive ELISA test, but it was not detected in any of the plasma samples by this method. The levels of C-reactive protein (CRP) were elevated in 5 out 6 animals from day 1 post-challenge compared to pre-challenge baseline levels (Figure 4F).

#### 3.4.6. Additional Immunological Assessment Indicates a Lack of Adaptive or Memory Response to Infection

To assess for the T-cell memory response, single-cell suspensions of spleen homogenates and white blood cell cultures from animals euthanised on days 2, 4, or 21 were stimulated with SARS-CoV-2 spike peptides. On day 2 post-challenge, a strong response was observed in animal M1, suggesting that the splenic T-cells were specifically responding to the SARS-CoV-2 spike peptides (data not shown). However, there was no evidence of T-cell memory in either spleen cells or white blood cells of the other animals as determined by a lack of response to the peptides. Antibody responses were assessed in marmosets that were euthanised at the end of the study on day 21 post-challenge. IgM and IgG antibodies to SARS-CoV-2 spike protein were not detected by ELISA. Neutralising antibodies to SARS-CoV-2 VIC01 were not detected in a plaque reduction neutralization test (PRNT).

### 3.5. Minimal Histopathological Changes Were Observed in Respiratory Tissues

Histopathological analysis was carried out on respiratory and non-respiratory tissue samples collected on days 2, 4, and 21 post-challenge. The majority of the surface of lung tissue did not show any remarkable changes and was within normal limits (Figure 5A); however limited, minimal lesions were observed in the lungs of all animals. These lesions comprised of a few foci of interstitial inflammation (Figure 5B) and minimal inflammatory cell infiltration within the presence of occasional scattered multinucleated giant cells (Figure 5C). No significant lesions were observed in the upper respiratory tract as evidenced in the trachea (Figure 5D) and nasal cavity (Figure 5E). No significant lesions were found in extra pulmonary tissues that were studied; however, background findings could be identified in the liver, with small foci of extramedullary hemopoiesis (EMH) and vacuolation of hepatocytes, related to glycogen storage in all animals (Figure 5F). Minimal to mild infiltration of mononuclear inflammatory cells was also observed within the kidney cortex, mainly in animals M1 and M3 (Figure 5G and data not shown). EMH was frequently observed in the spleen, with abundant megakaryocytes within the splenic red pulp (Figure 5H). The presence of viral RNA was assessed using RNAScope, but no SARS-CoV-2 RNA was detected in any of the tissues studied from all animals (Figure 6A–C). Strong positive staining was observed in all the positive control sections (Appendix A).

### 3.6. There Was a Lack of Expression of the ACE2 Receptor in Marmoset Lower Respiratory Tissues, However TMPRSS2 Was Ubiquitously Expressed in Multiple Tissues

The expression patterns of two key host factors, ACE2 and TMPRSS2, were assessed in respiratory and non-respiratory tissues. For ACE2, intense staining was observed in the apical border of the small intestine absorptive epithelium (Figure 7A). Mild staining was observed in glandular epithelial cells from the small intestine. The staining in the large intestine was minimal (Figure 7B) and was almost absent in the trachea, lung, and nasal cavity (Figure 7C–E), compared to a moderate positive reaction in these tissues from rhesus macaque positive control sections (Appendix A). There was moderate staining in the glandular epithelium within the olfactory mucosa area of the nasal cavity (Figure 7F); however, staining was absent in the epithelium lining and nerve terminations. Intense staining was also observed within the tubular epithelial cells of the kidney and was mostly within the apical border (Figure 7G). Mild to moderate staining was observed in immune cells present in the liver (mostly within EMH foci) (Figure 7H) and there was no staining in the spleen, thymus, or brain (Figure 7I). For TMPRSS2, a strong reaction was observed in the small and large intestine (Figure 8A,B), mostly within immune cells within the mucosa, but also in the absorptive epithelial cells and smooth muscle cells. Positive staining was also observed within smooth muscle layers of the oesophagus (Figure 8C). Strong staining was observed in some airway epithelial cells in the lung, with less intensity in the alveoli and inter alveolar septa (Figure 8D). In the larger airways and upper respiratory tract (bronchi and trachea), moderate staining was also observed within the respiratory epithelium and lamina propria (Figure 8E,F). Levels of TMPRSS2 staining were comparable to tissue samples from experimentally infected rhesus macaques used as positive controls (Appendix A).

## 4. Discussion

This study is the first example of exposure of marmosets to aerosolized SARS-CoV-2 to deliver the virus directly to the lower respiratory tract. Prior to the challenge, optimization studies were performed to determine the appropriate aerosolization conditions for SARS-CoV-2. These studies indicated that there was a reproducible, proportional relationship between the starting titre of the virus in the Collison nebuliser and the titre of the virus recovered in the impinger. The Spray Factor was not related to the concentration of the virus recovered in the impinger, which indicated that the efficiency of the aerosol delivery system was not compromised by other compounding factors. Changes in the relative humidity were assessed to determine the optimal viral recovery following aerosolization. Spray Factor suggested better recovery at low relative humidity than medium or high relative humidity.

In this study, key features of COVID-19 observed in human disease such as fever, respiratory dysfunction, virus replication, histopathological changes, and immunological responses to infection were studied in the marmoset. Marmosets challenged with SARS-CoV-2 by the aerosol route did not develop overt signs of clinical disease; however, there was some evidence for weight loss during the initial phase following the challenge. In addition, although the animals did not develop a typical fever, a transient disruption to the diurnal temperature profiles of three out of six animals was observed.

Infectious virus could not be recovered from marmoset tissues at post-mortem, and the presence of viral RNA could only be detected in the lungs of animals on days 2 and 4 post-challenge by RT-PCR. Viral RNA was detected in nasal and/or throat swabs on day 1 post-challenge only, with the exception of animal M6 where viral RNA was still detectable in nasal swabs until day 7. The lack of evidence for virus replication in the upper airways and the lungs and dissemination of virus to other tissues would suggest that the challenge input virus was being detected in these assays. Similar findings have also been reported by Lu et al. [18]. A sub-genomic PCR would be required to confirm this, which was outside of the scope of this study.

Despite the lack of clinical disease, marmosets developed a profound innate immune response following SARS-CoV-2 challenge with prolonged disruptions to the proportions of circulating neutrophils and T cells. Elevated proportions of neutrophils were sustained for the 21-day duration of study, with a slight decline in levels on day 7. A similar profile was also observed in marmosets challenged with MERS-CoV reaching a similarly high level of approximately 60% of the PMBC’s [16]. However, in contrast to infection with SARS-CoV-2, the neutrophil proportions in marmosets challenged with MERS-CoV declined and returned to baseline levels by day 7. Infection with MERS-CoV also resulted in activation of the neutrophils, which were CD54^+^ and CD80^+^, whereas there was no change in the activation markers following infection with SAR-CoV-2, apart from the HLA-DR marker. A reduction in neutrophil HLA-DR expression was observed in some animals infected with SARS-CoV-2 and was suggestive of ill health in these marmosets. The expression of HLA-DR on neutrophils is not observed in humans, but it has been linked with a number of bacterial and viral diseases in the marmoset including the febrile response in Q Fever, melioidosis, and MERS [16,26,27]. The lack of expression of the other markers that would typically occur in response to a developing infection was not observed, indicating that the infection was cleared quickly and a maturing immune response did not develop.

The sustained increase in neutrophil proportions during an infection is unusual as neutrophils are considered to be short-lived cells (less than 1 day). Persistence of activated neutrophils have been reported [28] and sustained levels in COVID-19 patients is associated with severe disease [29]. In these marmosets, the expansion of the neutrophil population was associated with the initial challenge, and this pool of neutrophils in other marmoset infections would usually be exhausted in an attempt to control a severe infection [30] or return to normal levels for mild infections [27]. Following the SARS-CoV-2 challenge, however, the virus was cleared quickly, therefore the neutrophil populations were not expended but persisted for the duration of the study. The apparent reduction in T-cells starting from day 1 post-challenge is difficult to explain, as there was no evidence for migration of T-cells into the lung tissue with only minimal inflammation observed by histopathology. However, a later lymphopenia is characteristic of COVID-19 in humans and is linked to severe disease [31].

Severe COVID-19 infection in humans is associated with immune dysfunction including the cytokine storm [32] and macrophage activation syndrome [33]. Assessing these parameters in the marmoset did not indicate any dysfunction; despite an increase in the levels of monocytes there was minimal systemic activation of the cells. However, levels of C-reactive protein (CRP) were elevated in five out of six animals from day 1 post-challenge compared to pre-challenge baseline levels, but these levels were below diagnostically significant levels described in human COVID-19 cases [34]. There was no evidence of a developing adaptive immune response (indicated by either T-cell recall assays, identification of memory cells, or antibody production). This provides further evidence that despite a high dose challenge, the virus was cleared before the infection could establish and the adaptive immune response could develop. There was no evidence of pathology associated with SARS-CoV-2 infection in the respiratory tissues or major organs at the macroscopic and microscopic level. Overall, no significant histopathological changes were observed in fixed tissues at any time point. This coincided with the absence of viral RNA using a very sensitive and specific RNAScope ISH technique to target the *S-gene*. This method has also been successfully used for tissues from rhesus and cynomolgus macaques [35], ferrets [36], and hamsters [37], and even though the RNA probe used in the ISH RNAScope analysis was not targeting the replicating virus (sense probe), these studies have found a good correlation between infectious virus and presence of ISH positive staining. In addition, a red chromogen in the RNAScope ISH technique was used to avoid any misinterpretation of small amounts of pigment present in multiple tissues. The absence of viral RNA using RNAScope is at odds with the RT-PCR data where viral RNA was still detectable in the lungs of marmosets on days 2 and 4 post-challenge and in nasal and/or throat swab samples, and this may be due to a higher limit of detection for the RNAScope method as there is no amplification phase, compared to the RT-PCR method.

It cannot be excluded that these observations could be attributed to components of the virus diluent used for challenge or the frequent anaesthesia for bio-sampling. The use of mock-infected animals was outside the scope of this pilot study, however future studies could include these controls to address this.

The entry of SARS-CoV-2 into host cells depends on binding of the viral spike (S) protein to the ACE2 host receptor and the priming of the viral S by host TMPRSS2 [38]. To further understand the interactions between the ACE2 receptor and S protein, Damas et al. have conducted studies to assess binding affinities and predict species susceptibility to SARS-CoV-2 infection [39]. Their results indicated that 21 of the 24 amino acids in the binding region of the marmoset ACE2 were identical to the human ACE2 binding region, predicting medium susceptibility to SARS-CoV-2. This would suggest that the marmoset may be suitable to model COVID-19. In this study, protocols were established for the immunohistochemical detection of ACE2 and TMPRSS2 in marmoset tissues to explore the abundance and distribution of ACE2 and TMPRSS2 in respiratory and non-respiratory tissues. ACE2 staining was strong in the intestinal absorptive epithelium as well as kidney tubular epithelial cells; however, the respiratory tissues showed almost non-existent staining overall. The absence of the ACE2 receptor is supportive of the lack of evidence for virus replication in the lung and other respiratory tissues that has been observed in this study, even though TMPRSS2 was present in multiple cell populations in tissues. More recent evidence has indicated that even though the marmoset ACE2 orthologue differs from the human ACE2 receptor by only four amino acids [39], the H41 and E42 residues are critical for SARS-CoV-2 infectivity [40]. This would explain the findings of this study. Furthermore, humanization of the marmoset ACE2 by mutating the orthologue at positions 41 and 42 into Y and Q, respectively, was able to restore S1 binding and SARS-CoV-2 infectivity.

In summary, marmosets did not develop overt disease when challenged with SARS-CoV-2 by the aerosol route, and there was limited evidence of viral replication or pathology associated with infection. This data is in agreement with other research groups who have reported only mild disease in marmosets challenged with SARS-CoV-2 by alternative routes such as intranasal or combination of intranasal/intratracheal/oral/ocular [17,18]. However, further studies to optimise binding of SARS-CoV-2 to the ACE2 receptor could be considered to explore the common marmoset as an alternative non-human primate species to model COVID-19.

## Figures and Tables

**Figure 1 viruses-14-01580-f001:**
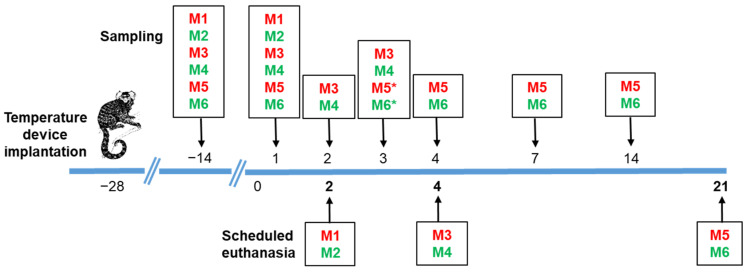
Schedule of in-life and post-mortem sampling. Animals (red—female; green—male) were surgically implanted with a Remo 201 device (EMMS, Bordon, UK) and allowed 4 weeks to recover prior to challenge with SARS-CoV-2. Baseline blood samples were collected 2 weeks prior to challenge. On days 1, 2, 3, 4, 7, 14, and 21 post-challenge, groups of animals were anaesthetized and blood, throat, and nasal swabs were collected to assess viremia, immunological parameters, and virus shedding (* note that blood samples were not collected from M5 and M6 on day 3 post-challenge). Pairs of marmosets were euthanised at day 2, 4, and at the end of the study on day 21 post-challenge and blood and tissues were collected at post mortem for further analyses.

**Figure 2 viruses-14-01580-f002:**
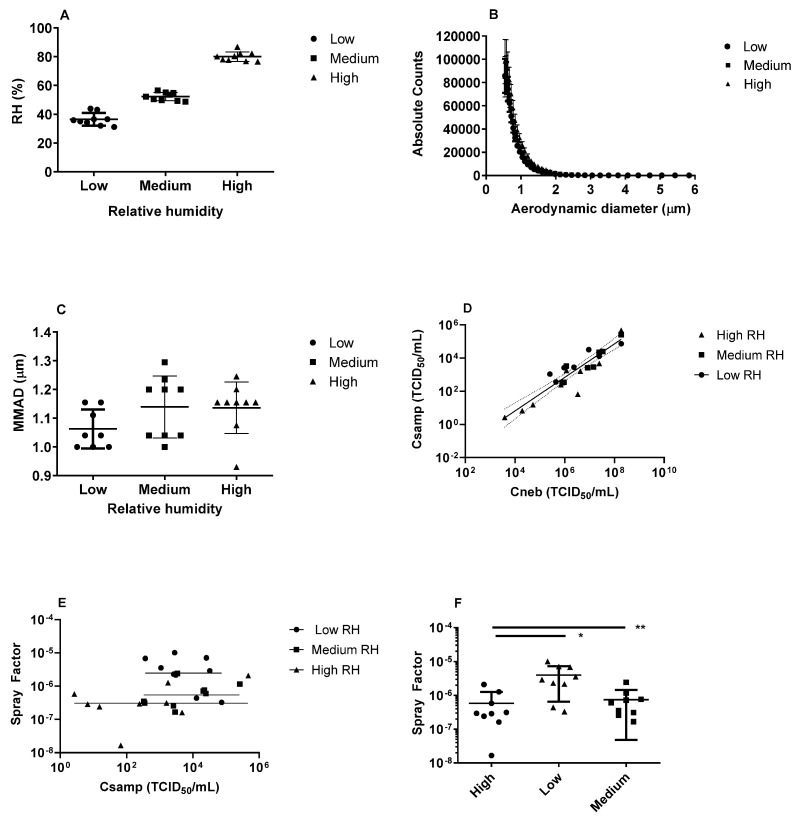
Optimization of the aerosolization of SARS-CoV-2. (**A**) The relative humidity conditions explored; (**B**) Particle size distribution at different relative humidities; (**C**) Mass median aerodynamic diameter (MMAD) at each relative humidity; (**D**) Comparison of nebuliser (C_neb_) and impinger (C_samp_) concentrations at low, medium, and high relative humidity. Individual data points are presented with calculated linear regression line and the 95% confidence interval; (**E**) Comparison of the Spray Factor and impinger (C_samp_) concentration at low, medium, and high relative humidity. Individual data points are presented with calculated linear regression line for each relative humidity subgroup; (**F**) Comparison of the Spray Factor at low, medium, and high relative humidity. Data is presented as the mean and SD where * *p* = 0.0266 and ** *p* = 0.0022.

**Figure 3 viruses-14-01580-f003:**
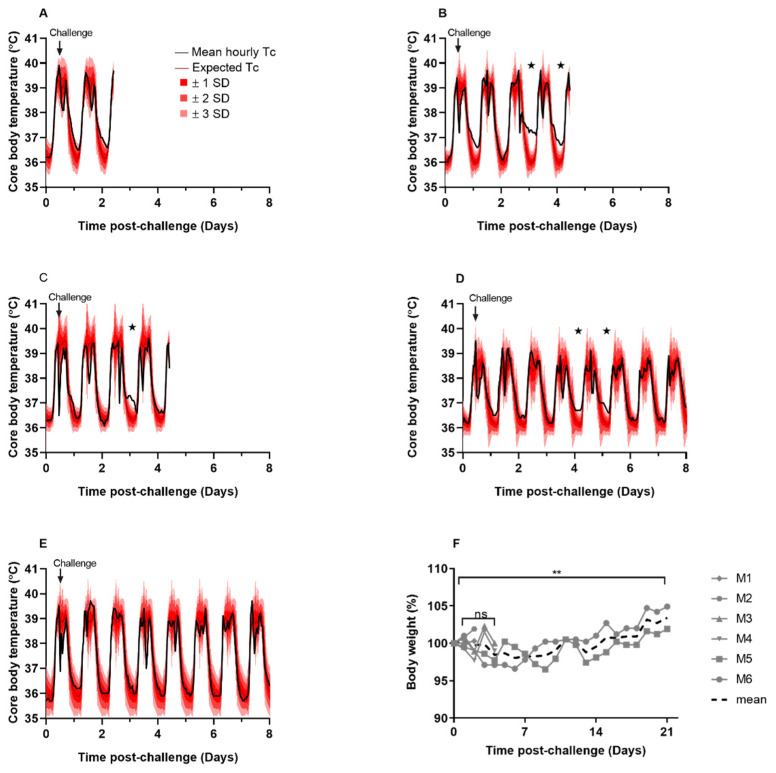
Clinical observations in marmosets challenged with SARS-CoV-2 by the aerosol route. Remote telemetry was used to monitor core body temperature (Tc) in marmosets. Normal diurnal temperature ranges between 38–39 °C during the day and 36–37 °C at night. The mean hourly temperature (black line), the expected temperature (using baseline data for each individual animal; dark red line), and the expected temperature ±1 SD (bright red shaded area), ±2 SD (red shaded area), and ±3 SD (pink shaded area) are shown for (**A**) M1, (**B**) M3, (**C**) M4, (**D**) M5, and (**E**) M6 from days 0 to 8 post-challenge. ★ Indicates Tc > 3 SD than the expected Tc (note that there is no Tc data for M2 due to failure of implant). (**F**) Animals were weighed prior to the challenge to determine baseline weights and daily throughout the study. Data is presented as the percentage body weight change for each animal, or the mean of all animals. A mixed-effects model was used to determine statistical difference in weight change with time where ns—not significant and ** *p* = 0.007.

**Figure 4 viruses-14-01580-f004:**
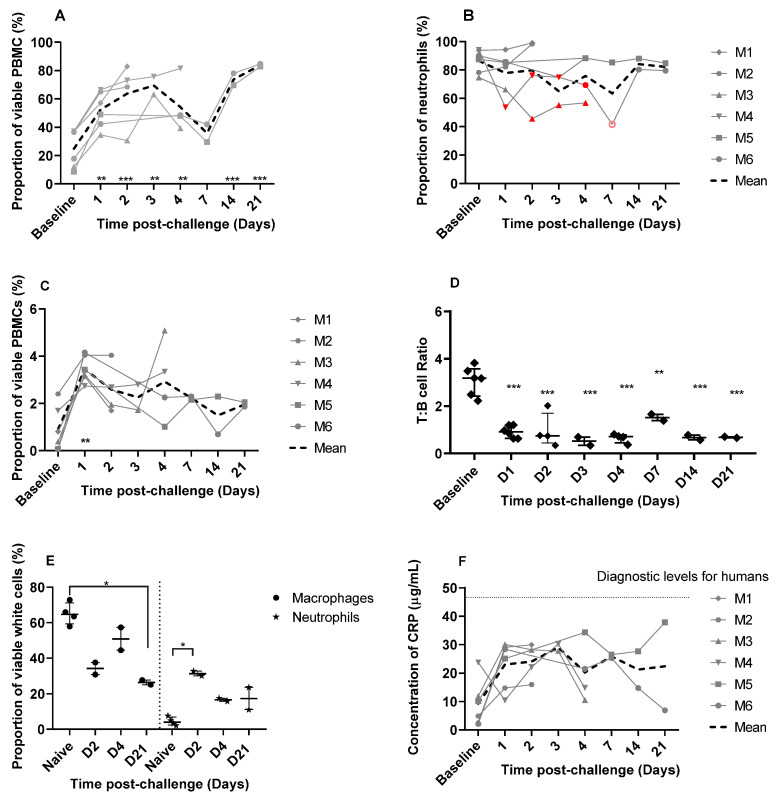
The immunological response in marmosets following aerosol challenge with SARS-CoV-2 VIC01. (**A**) The proportion of neutrophils in PBMCs from blood collected prior to, and post-challenge defined by size, granularity, and expression of both CD11c and CD14. (**B**) HLA-DR expression in circulating neutrophils in marmosets following SARS-CoV-2 challenge. Filled red symbols denote a reduction in HLA-DR expression (defined as 10% below baseline for each individual animal determined from pre-challenge samples); pink open circle denotes a poorly stained sample on day 7 for animal M6 and this data therefore should be treated with caution. (**C**) The proportion of monocytes in PBMCs from blood collected prior to and post-challenge. The T:B cell ratio (**D**) in PBMCs from blood collected prior to challenge and on days 1, 2, 3, 4, 7, 14, and 21 post-challenge were defined by size, granularity, and expression of CD3^+^:CD20^+^. (**E**) Proportion of macrophages and neutrophils in marmoset lung following SARS-CoV-2 challenge (n = 2 per time point). Samples from naïve animals (n = 4) were processed during the study to provide resting levels. (**F**) Quantification of CRP in marmoset plasma detected by ELISA in marmoset plasma samples. Data is presented for each animal. The COVID diagnostic level is the threshold of diagnostically significant levels of CRP described in human COVID-19 cases. In all figures, data is presented for each animal and the normal ranges were defined by the pre-challenge values. Statistically significant differences from pre-challenge values were determined by one-way ANOVA where * *p* < 0.05, ** *p* < 0.01 and *** *p* < 0.001.

**Figure 5 viruses-14-01580-f005:**
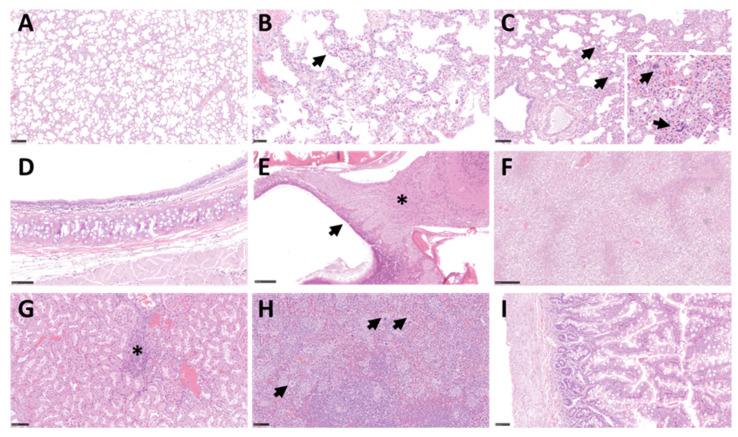
Histopathological findings in the respiratory and non-respiratory tissues following SARS-CoV-2 challenge. H&E-stained lung tissue from animal M2 (**A**) shows minimal changes and within normal limits; from animal M3 (Bar = 250 µm) (**B**) shows minimal septal inflammatory cell infiltration (arrow), and animal M5 (Bar = 250 µm) (**C**) shows scattered foci of extramedullary hematopoiesis within the lung parenchyma (Bar = 100 µm) (arrows and insert). No significant lesions were observed in H&E-stained trachea from animal M2 (**D**) and in the nasal cavity of animal M5 (Bar = 250 µm) (**E**) with normal histological structure in the olfactory mucosa (arrow) and the olfactory nerves and bulb (*). H&E-stained liver tissue section from animal M1 (Bar = 250 µm) (**F**) shows moderate vacuolation of hepatocytes; kidney tissue section from animal M1 (Bar = 100 µm) (**G**) shows mononuclear inflammatory infiltrates within the cortex (*); and spleen tissue section from M3 (Bar = 100 µm) (**H**) shows abundant megakaryocytes within the red pulp (arrows); (**I**) shows small intestine tissue section from M4 with no significant changes (Bar = 100 µm).

**Figure 6 viruses-14-01580-f006:**
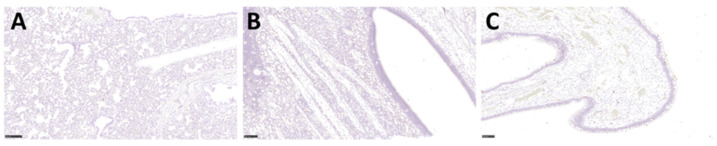
Representative images of ISH of respiratory tissues from animal M2 by RNAScope. Sections of lung (Bar = 250 µm) (**A**), nasal olfactory cavity (Bar = 100 µm) (**B**), and respiratory epithelium (Bar = 100 µm) (**C**) were stained for the presence of the SARS-CoV-2 S gene. No positive staining was detected within the lung, respiratory epithelium, olfactory epithelium, glands, or nerve terminations.

**Figure 7 viruses-14-01580-f007:**
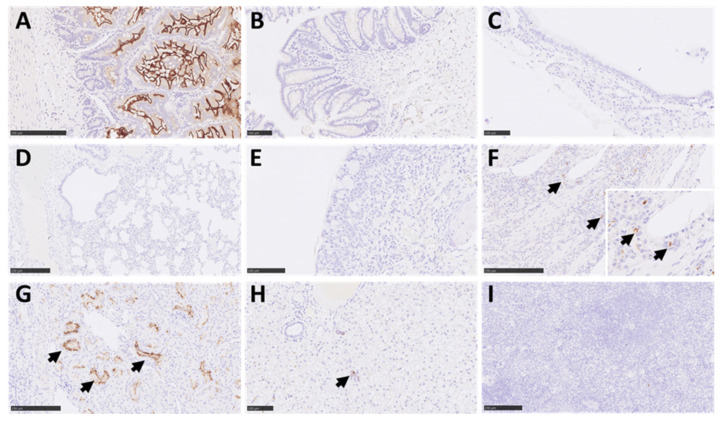
Representative images of IHC for ACE2 receptor in marmoset tissues. Positive staining was observed in the small intestine (Bar = 250 µm) (**A**), mostly in the apical border of absorptive epithelial cells and at a low level in some epithelial cells of glands. Minimal staining in epithelial cells from the large intestine (Bar = 100 µm) (**B**). The positive staining in the trachea (Bar = 100 µm) (**C**), lung (Bar = 250 µm) (**D**), and respiratory nasal mucosa (Bar = 100 µm) (**E**) was almost non-existent. However, moderate positive staining was observed in glandular epithelium within the olfactory mucosa area of the nasal cavity (arrows and insert; Bar = 250 µm) (**F**). To the contrary, there was no staining in the epithelium lining or nerve terminations. Intense staining was observed in the kidney in the apical border of kidney tubular epithelial cells (arrows; Bar = 250 µm) (**G**) and moderate reaction in immune cells within the liver (Bar = 100 µm; arrows, normally extramedullary haematopoiesis) (**H**). No reaction was observed in the spleen (Bar = 250 µm) (**I**).

**Figure 8 viruses-14-01580-f008:**
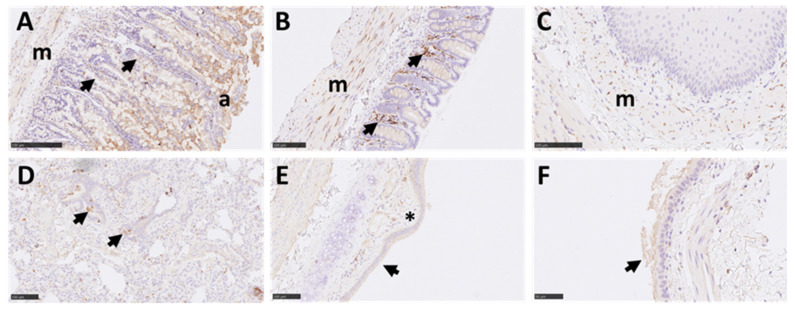
Representative images of IHC for TMPRSS2 in marmoset tissues. Strong reaction was observed in the small (Bar = 250 µm) (**A**) and large intestine (Bar = 100 µm) (**B**), mostly within immune cells within the mucosa (arrows), but also absorptive epithelial cells (a), smooth muscle (m), and few blood vessels. The staining within smooth muscle (m) can also be observed within the oesophagus (Bar = 100 µm) (**C**). Strong positive staining was also observed in some airway epithelial cells in the lung parenchyma (Bar = 100 µm) (**D**) (arrows), with less intensity in the alveoli and septa. In larger airways and upper respiratory tract (bronchi (Bar = 100 µm) (**E**) and trachea (Bar = 250 µm) (**F**)), moderate positivity can also be observed within the respiratory epithelium (arrows) and lamina propria (*).

## Data Availability

Not applicable.

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
