# Peer review of "Histopathological and Immunological Findings in the Common Marmoset Following Exposure to Aerosolized SARS-CoV-2"

_viruses, 2022, doi:10.3390/v14071580_

Round 1
Reviewer 1 Report
The authors describe the challenge of common marmosets with SARS-CoV-2 by the aerosol route with the aim of exploring their potential as an animal model. Unfortunately, the animals were not infected. The authors state (line 671) that the lack of infection of marmosets can be explained by amino acid changes in the ACE2 receptor. It would seem appropriate that this (and lack of respiratory tract expression of ACE2) is more prominently presented in the manuscript.
A major omission from this study is the lack of control animals challenged with mock aerosolised inoculum. It is unproven what effect the virus diluent (containing FBS) may have had, although the effects seem likely to have been minimal.
It is not clear why only proportions of white blood cells were measured. It would have been more informative to know actual numbers of WBC subtypes following virus challenge. I am not sure that the percentage of each subtype tells us anything useful; though the shift in T:B ratio is interesting it is not possible to assign this to an increase in circulating B cell numbers or decrease in T cells.
Lines 441 – 449. Why are these cells referred to as both monocytes and macrophages? Please be sure the writing is clear where increases refer to a proportion of cells rather than absolute cell counts.
Line 450 Lymphopenia was observed despite the lack of T-cell activation. It is not possible to determine this due to the lack of absolute cell counts. Similar observation regarding neutrophils.
Line 494. In the absence of any mock challenged animals how is it possible to assign any of the observed (minimal) changes to virus challenge? e.g. lesions shown in 5b.
Line 606. This statement has not been shown in the results as there are no absolute blood counts measured: “Despite the lack of clinical disease, marmosets developed a profound innate immune response following SARS-CoV-2 challenge with evidence of neutrophilia and lymphopenia”
Line 678. “SARS-CoV-3” I hope this is a typo!
Author Response
Thank you for your review. Where possible we have made improvements to the manuscript based on your review. We hope that we have been able to address your comments and that the manuscript can be accepted for publication. Please see specific responses to each of your points below:
- The authors state (line 671) that the lack of infection of marmosets can be explained by amino acid changes in the ACE2 receptor. It would seem appropriate that this (and lack of respiratory tract expression of ACE2) is more prominently presented in the manuscript.
As suggested this has been discussed more thoroughly in the discussion section of manuscript.
- A major omission from this study is the lack of control animals challenged with mock aerosolised inoculum. It is unproven what effect the virus diluent (containing FBS) may have had, although the effects seem likely to have been minimal.
Under UK Home Office Project Licence, we are expected to conduct initial susceptibility studies using the fewest number of animals, in the spirit of the 3Rs. Therefore, we designed this study to obtain maximum information using just 6 animals challenged with SARS-CoV-2, inclusion of diluent only challenged animals was outside the scope of this study. These could be included in follow on experiments, if further studies were appropriate. We have observed only limited evidence of infection in these animals, we believe that we have been clear and transparent in the manuscript to show that these observations may or may not be associated with virus challenge. We have included a short paragraph in the discussion to acknowledge this.
- It is not clear why only proportions of white blood cells were measured. It would have been more informative to know actual numbers of WBC subtypes following virus challenge. I am not sure that the percentage of each subtype tells us anything useful; though the shift in T:B ratio is interesting it is not possible to assign this to an increase in circulating B cell numbers or decrease in T cells.
Due to small size of a marmoset, we are limited to small volumes of whole blood collected at each time point. With such small volumes, it can be difficult to accurately measure the quantity of blood withdrawn at each occasion and furthermore the sample is diluted in the vial’s anticoagulant quickly as marmoset blood clots readily. The sample size obviously has a profound effect on cell density, which is further compounded by the multiple wash/processing steps that are manually performed within the containment facility. These factors have less impact on cell proportions, consequently presenting the data in this way is in our opinion more reliable, an essential factor given our group sizes. However we have now included some total count data to help clarify the observed changes and added a statement into the methods to explain why statistical significance is only calculated for cell proportions.
- Lines 441 – 449. Why are these cells referred to as both monocytes and macrophages? Please be sure the writing is clear where increases refer to a proportion of cells rather than absolute cell counts.
Macrophages was used in error; we have now corrected this in the manuscript. We have included the extra statement in the statistical methods to clarify why we have chosen to present proportions as opposed to cell counts.
- Line 450 Lymphopenia was observed despite the lack of T-cell activation. It is not possible to determine this due to the lack of absolute cell counts. Similar observation regarding neutrophils.
We have removed the terms lymphopenia and neutrophilia in order not to over-state our data, and included average cell densities to support our claim that the change in T:B ratio is the result in depression in T cells
- Line 494. In the absence of any mock challenged animals how is it possible to assign any of the observed (minimal) changes to virus challenge? e.g. lesions shown in 5b.
Our intention was to be very transparent showing these animals can show “background” lesions or minimal to mild lesions that can be or not associated to the virus. In our case, we cannot see virus in the lower respiratory tract so the presence of lesions is open to interpretation. We believe that this is clear in our manuscript. Indeed mock infected animals would be useful to assign whether these observations were due to virus challenge, however as stated in Q2, this was outside the scope of this pilot study.
- Line 606. This statement has not been shown in the results as there are no absolute blood counts measured: “Despite the lack of clinical disease, marmosets developed a profound innate immune response following SARS-CoV-2 challenge with evidence of neutrophilia and lymphopenia”
We have changed the text to read as not to over-state our data: “Despite the lack of clinical disease, marmosets developed a profound innate immune response following SARS-CoV-2 challenge with a prolonged disruptions to the proportions of circulating neutrophils and T cells.”
- Line 678. “SARS-CoV-3” I hope this is a typo!
Apologies, this is indeed a typo and has been corrected in the manuscript
Reviewer 2 Report
The authors characterize infection of marmoset with aerosolized SARS-CoV-2. The results of the experiment demonstrate that there were no clinical signs of disease and limited viral RNA detected by real-time RT-PCR.
Minor comments
Line 389 Section
Could the authors add the CT values of the real-time PCR to the data. This is important to show the level of viral RNA isolated.
Author Response
Thank you for your review. We have made improvements to the manuscript based on your review. We hope that we have been able to address your comment and that the manuscript can be accepted for publication. Please see specific response to your point below:
Line 389 Section
Could the authors add the CT values of the real-time PCR to the data. This is important to show the level of viral RNA isolated.
On reflection, we agree that it is useful to the reader to include this information, particular as in most cases high Ct values were obtained from those samples that we have reported as positive for viral RNA. We have amended the results section to include the Ct values where appropriate.
Reviewer 3 Report
Dear all,
the study Ireland et al. thoroughly descirbes findings in marmosets after exp. SARS CoV-2 infection and the authors show clear results, which are well illustrated, very informative and helpful for current subject. Furthermore, the mansucript is well written and organised. Thank you so much.
However, few suggestions:
1.) please go through all consumables and make sure, the audience is able to purchase the same products easily, e.g. it is a bit misleading that the ACE2 and TMPRSS2 antibodies are from Biotechne, as the product code clearly states, the product is from NovusBio. please check all consumables accordingly.
2.) in figure 5 C, the lung, the so-called "multinucleated cells" are extramedullary haematopoesis - I assume, which is often seen in systemic diseases and represents a common feature for Covid-19 (human) patients as well as other animal species after SARS CoV-2 infection. Please correct that - as multinucleated giant cells might mislead the reader and indicate a kind of granulomatous inflammation, which is not the case. Please clarify.
3.) please comment on how the histological changes (HE and also ISH -virus detection) have been quantified in the animals and state in the M&M.
4.) Am I right that no "mock" animals have been included? the data are not compared to control animals to proof that frequent anesthesia might have an impact on the immune system as observed, please clarify this and highlight this in the manuscript and/or provide data from other studies.
5.) It is surprising that no positive signal is detected in the upper resp. tract by ISH at 1 dpi, if the swabs are positive by PCR (to my experience) there is a signal also in the nose/turbinates, sometimes only in few immune cells and debris by RNA scope. Please comment on that.
6.) can you please provide evidence that RNA integrity of the virus for a successful ISH after decalcification is maintained? and i am not talking about a reference gene like ubiquitin after decalcification.
7.) can you please provide a graphical representation of the ACE2 and TMPRSS2 expression in the resp. tract - in addition to the photomicrographs for an easier appreciation of the dataset by the reader? how was this quantified?
Author Response
Thank you for your very kind review. Where possible we have made improvements to the manuscript based on your review. We hope that we have been able to address your comments and that the manuscript can be accepted for publication. Please see specific responses to each of your points below:
1.) Please go through all consumables and make sure, the audience is able to purchase the same products easily, e.g. it is a bit misleading that the ACE2 and TMPRSS2 antibodies are from Biotechne, as the product code clearly states, the product is from NovusBio. Please check all consumables accordingly.
We have gone through the M+M and made changes to ensure completeness.
2.) In figure 5 C, the lung, the so-called "multinucleated cells" are extramedullary haematopoiesis - I assume, which is often seen in systemic diseases and represents a common feature for Covid-19 (human) patients as well as other animal species after SARS CoV-2 infection. Please correct that - as multinucleated giant cells might mislead the reader and indicate a kind of granulomatous inflammation, which is not the case. Please clarify.
We agree with the reviewer. These cells are typically observed in several organs in this species (more often in young animals” as foci of extramedullary haematopoiesis). We have changed the text from “multinucleated cells” to “foci of extramedullary haematopoiesis”.
3.) Please comment on how the histological changes (HE and also ISH -virus detection) have been quantified in the animals and state in the M&M.
The histopathological changes observed in the different organs were just minimal or mild and we have not used any scoring system as in previous reports by our group (e.g. Salguero et al. 2021 Nat Comms or Dowall et al. 2021 Viruses MDPI) where lesions induced by the virus were evident and ranging from minimal to severe (rhesus and cynomolgus macaques or hamsters). We could not find any positive staining with RNAScope ISH so no quantification was carried out.
4.) Am I right that no "mock" animals have been included? The data are not compared to control animals to proof that frequent anaesthesia might have an impact on the immune system as observed, please clarify this and highlight this in the manuscript and/or provide data from other studies.
That is correct, we did not include “mock” infected animals in this initial pilot study but we have been able to compare some of findings to our previous studies in marmosets challenged with MERS-CoV. The frequency of anaesthesia does have the potential to impact the immune response and cannot be excluded during the earlier time points, however we also observed a prolonged immune response until the end of the study when animals were anaesthetised weekly. We have included a comment about the potential for anaesthesia to impact the immune responses, but we do not have any further data to prove or disprove this. We have included a short paragraph in the discussion to acknowledge this.
5.) It is surprising that no positive signal is detected in the upper resp. tract by ISH at 1 dpi, if the swabs are positive by PCR (to my experience) there is a signal also in the nose/turbinates, sometimes only in few immune cells and debris by RNA scope. Please comment on that.
In our experience with other animal models, we do detect some ISH positive staining at 1dpc in the nasal cavity after IN inoculation, but it is not observed in all animals even with positive PCR results. The absence of cell debris/immune cells in most of the nasal turbinates could be due to a washing-off effect during sampling and processing. No ISH +ve staining was observed in epithelial cells as we have described in other animals models (Ryan at al, 2021 Nat Comms for ferrets or Dowall et al. 2021 for hamsters).
6.) Can you please provide evidence that RNA integrity of the virus for a successful ISH after decalcification is maintained? and i am not talking about a reference gene like ubiquitin after decalcification.
SARS-CoV-2 positive hamster nasal cavities were processed in parallel to demonstrate that the RNA integrity was not compromised during the decalcification. We have included a representative image as an additional panel in Figure Supplementary Figure S1. We have also used the same decalcifying method for many other papers showing excellent virus RNA preservation for the RNAScope technique (e.g. ferrets in Ryan et al., 2021 Nat Comms or Bewley et al. 2021 Sci Advances; and hamsters in Huo et al. 2021 Nat Comms or Dowall et al. MDPI viruses)
7.) Can you please provide a graphical representation of the ACE2 and TMPRSS2 expression in the resp. tract - in addition to the photomicrographs for an easier appreciation of the dataset by the reader? How was this quantified?
We wanted to demonstrate the lack of the ACE2 receptor in the respiratory tissues and the ubiquitous expression of TMPRSS2 throughout tissues, and we feel that the images support these statements without the requirement to quantify levels of expression.
Round 2
Reviewer 3 Report
Dear all,
thank you for your reply.
re my initial comment 1.) Please go through all consumables and make sure, the audience is able to purchase the same products easily, e.g. it is a bit misleading that the ACE2 and TMPRSS2 antibodies are from Biotechne, as the product code clearly states, the product is from NovusBio. Please check all consumables accordingly.
We have gone through the M+M and made changes to ensure completeness.
As far as I can see it, you changed the company for the ACE2 ab, and not for the TMPRSS2. Furthermore, the majority of the other consumables in the manuscript lack product codes, which doesn't allow others to reproduce your work. I would like to mention again as in my first comment: Please check all consumables accordingly and add product codes and the company.
Thank you so much.
Author Response
Dear reviewer
Thank you for taking the time to review the revised version of the manuscript and for your further comments to ensure that product codes are included for materials and consumables. We have made these changes and feel that this added information would now allow others to reproduce our studies more easily. Thank you. We hope that we have addressed your comments to your satisfaction and that the manuscript is acceptable for publication.